# Working with Children with Autism Undergoing Health-Care Assessments in a Day Hospital Setting: A Perspective from the Health-Care Professionals

**DOI:** 10.3390/children10030476

**Published:** 2023-02-27

**Authors:** Chiara Davico, Daniele Marcotulli, Elisa Succi, Carlotta Canavese, Ancuta F. Bodea, Mariacristina Pellegrino, Enrica Cuffari, Valentina F. Cudia, Barbara Svevi, Federico Amianto, Federica Ricci, Benedetto Vitiello

**Affiliations:** 1Section of Child and Adolescent Neuropsychiatry, Department of Public Health and Pediatric Sciences, University of Turin, 10100 Turin, Italy; 2Department of Neuroscience, University of Turin, 10100 Turin, Italy

**Keywords:** autism, health-care professionals, EEG, medical testing, hospital

## Abstract

Background: Hospitals can be especially stressful for children with autism spectrum disorder (ASD) due to the communication and social skills deficits, lower capacity to adapt to disruption, and sensory hypersensitivity that are typical of these patients. Purpose: This study investigated how health-care professionals (HPs) experienced the clinical care and management of children with ASD undergoing medical testing in a day hospital setting, and assessed the rate of successful completion of laboratory tests and instrumental examinations. Methods: A cross-sectional questionnaire was administered to 45 HPs, inquiring about their experience in obtaining blood and urine tests, ECG, audiometry, and EEG from children with ASD. The clinical sample included 153 consecutively referred children with ASD (74.5% males, mean age 5.6 years) undergoing a medical diagnostic work-up as part of their diagnostic evaluation. The success rate of completing the various assessments was examined. Results: HPs identified aggressive behavior and communication deficits as the major challenges when providing care to children with ASD. The parents were seen as an important resource for managing the children. The completion rate of the laboratory tests and instrumental examinations was high (between 88.5% and 98.4% according to the specific type of examination). The lowest non-completion rate was found for the EEG (12.5%). Conclusions: Despite considerable challenges being reported by HPs in managing children with ASD, the scheduled assessments could be completed in the large majority of cases. Targeted approaches to preventing aggressive behaviors and obviating the communication barriers in children with ASD undergoing hospital exams are warranted.

## 1. Introduction

Autism spectrum disorder (ASD) is a heterogeneous neurodevelopmental condition characterized by early onset deficit in social communication and reciprocity, together with a restricted range of interests and repetitive behaviors [1]. The diagnosis of ASD has a population prevalence of about 1%, higher in males, with a median male to female ratio of 4.2 [2]. The ASD pathogenesis is not fully understood, but there is general consensus that complex interactions between multiple genes are mainly responsible for the disorder and that epigenetic factors and exposure to environmental modifiers influence its expression [1].

Autistic patients are more likely to develop both medical and psychiatric comorbidities than the general population, possibly due to shared etiopathogenetic mechanisms linked to different clinical phenotypes [3]. The most frequent psychiatric comorbidities of ASD are ADHD, anxiety disorders, irritability, obsessive-compulsive disorders, and mood disturbances, which can increase the need for medical assessments [4]. ASD also increases the risk of epilepsy, immunological disorders, gastrointestinal disorders, eating and feeding issues, dental problems, and sleep difficulties [5]. Complicated interactions between different body systems suggest a greater likelihood for individuals with one comorbidity to also present with others. For example, in children with sleep problems and ASD, gastrointestinal problems and seizures are twice as common [6]. Several guidelines recommend evaluating the presence of medical or genetic conditions on the basis of physical examination, individual characteristics, and clinical judgment both at the time of initial ASD diagnosis and as a part of ongoing care [7].

In further detail, in autistic patients, a blood test with analysis of blood count, erythrocyte sedimentation rate (ESR), C-reactive Protein (CRP), autoimmunity test, thyroid function tests, screening for anemia (iron, ferritin, transferrin, B12 vitamin, folic acid), electrolytes, pancreatic, and liver function, lipid profile, and blood sugar levels may be evaluated, especially in patients with eating and feeding disorders or with gastrointestinal symptoms (such as abdominal pain, diarrhea, or constipation). Restricted and repetitive patterns of behavior and associated sensitive alterations are also manifested in the way these patients feed themselves. In children with ASD, there is indeed a high prevalence of food selectivity for both the type and amount of food, abnormal mealtime habits, and symptoms of avoidant/restrictive food intake disorder (ARFID) [1,8]. This means that these patients can present not only alterations in body weight but also nutritional deficits, such as a lack of micronutrients and iron deficiency anemia. Medical assessments and procedures are often necessary to evaluate these feeding difficulties and identify deficits that can be managed and corrected to reduce negative impacts on physical health [9].

Various metabolic abnormalities have been shown in patients with ASD (e.g., disorders of amino acid metabolism and transport, organic acidurias, cholesterol biosynthesis defects, neurotransmitter disorders, and cerebral creatine deficiency syndromes). Prompt diagnosis of inborn errors of metabolism can increase the opportunity for early intervention and improvement [10,11].

Since about 1 in 10 individuals with ASD also have epilepsy [12], which is particularly associated with the presence of intellectual disability and dysmorphic features, electroencephalography (EEG) is frequently performed in patients with ASD if there is suspicion of epilepsy [10] or to monitor it. EEG is also recommended when a child with ASD has paroxysmal seizures or a regression of language has been observed [11].

Several guidelines suggest routine clinical genetic testing including fragile X syndrome testing and a chromosomal microarray to detect copy-number variants. Genomic microarrays are now indicated as a first-level diagnostic test for individuals with developmental delay, intellectual disability, multiple congenital anomalies, and ASD [13]. In some cases, genetic tests can identify possible causes of ASD or co-occurring medical conditions, thus helping in the development of more specific treatment plans [14]. Better identification of genetic causes of autism could increase the accuracy of genetic counseling for parents of children with ASD and other families considered at higher risk [15]. Expanded genetic counseling for the entire family is also necessary in view of a possible new pregnancy.

The assessment of a child with ASD or on suspicion of ASD should include formal audiologic evaluation [16]. Generally, behavioral audiometry is performed; alternatively, the auditory brainstem response or brainstem auditory evoked-response should be ordered. It is important to exclude a hearing impairment, since very often the first reason for referral in autistic toddlers is a concern about the child’s communication ability [17]. Some symptoms found in ASD, such as the lack of response to name calling or the tendency to social isolation may be due to a condition of hearing deficit.

Electrocardiographic (ECG) examination is often requested for children who are prescribed antipsychotics to rule out possible conduction abnormalities as shown by a heart rate-corrected QT interval (QTc) [18]. In fact, even though there are as yet no approved drugs to treat the core symptom of ASD, medications are often prescribed to children with ASD for control of behavioral problems. The drugs most commonly used in ASD are antipsychotics, antidepressants, and central stimulants. Some of these medications can cause QTc prolongation, which increases the risk of developing life-threatening ventricular arrhythmias.

For all these reasons, several guidelines include recommendations to evaluate the possible presence of medical and genetic co-existing conditions through physical examination, laboratory tests, and instrumental assessments both at the time of initial ASD diagnosis and as a part of ongoing management [15]. Blood and urinary tests including the evaluation of the most common metabolic abnormalities associated with autism, genetic tests for fragile X syndrome and chromosomal microarrays, EEG, audiometric evaluation, and ECG are usually performed [13,14].

The prevalence of ASD is steadily increasing, likely due to both actual increases in cases and improved diagnostic capabilities [2]; thus, the use of hospital services by these patients is also growing. Higher rates of health-care use are reported in ASD than in the general population. Children with ASD have more annual physician visits, more emergency room visits, and are more likely to be hospitalized [19]. There is also a higher burden of unmet health needs and lower satisfaction with the medical care received [20].

Medical tests and procedures conducted in hospital settings can be considered generally stressful to children, with a degree that is influenced by factors such as age, developmental stage, health status, and the type of test or procedure [21]. For example, MRI scans can be stressful for the need to stay still in an enclosed space and for the exposure to loud noises [22]. Children with ASD can be especially sensitive to the stressful stimuli that hospital procedures involve.

In general, children with ASD may exhibit extreme anxiety in interfacing with an unfamiliar situation and environment such as the hospital. In particular, the alterations in sensory perception, communication, and social deficits that are typical of the disorder make the hospital environment very difficult for these patients to tolerate and can even lead to a trigger of their behavioral problems [23]. Based on a literature review conducted by Strauss and colleagues in 2019, the hospital setting exacerbates challenging behaviors of children with ASD, and this interferes with the successful completion of the medical encounters [24]. Non-cooperation is reflected in a poor performance in meeting the requests of health-care encounters [25]. These difficulties increase the burden of care on caregivers, including parents and health-care professionals (HPs). Moreover, health-care providers report poor communication, rigidity of protocol, and lack of training about ASD as barriers to providing adequate care for these patients [26].

It is therefore important to understand the difficulties and communication issues related to ASD from the perspective of the HPs [3]. A recent review highlighted the perception of limited knowledge and self-efficacy of HPs when working with people with autism. The study also found that treating people with autism is felt as “uncomfortable” or “difficult” by a large proportion of HPs [27]. However, to our knowledge, no study has investigated the completion rates of the most common diagnostic procedures in children with autism and the perception of self-efficacy of the involved HPs.

This study aimed to assess the perception of HPs about the feasibility of conducting medical and laboratory tests among young children with ASD, and to examine the success rate in completing the scheduled tests. The possible association of patient characteristics, such as sex, age, and severity of ASD symptoms or of emotional dysregulation (ED), with the successful completion of EEG was also examined.

## 2. Materials and Methods

### 2.1. Design and Setting

We conducted a cross-sectional survey among HPs working at the Division of Child and Adolescent Neuropsychiatry of the University of Turin (Pediatric Hospital Regina Margherita). The day hospital is dedicated to the care of children and adolescents with neurological and psychiatric conditions and maintains a specific day of the week on which patients with ASD are admitted for laboratory or instrumental tests, which in most cases require blood drawing, urine collection, and EEG. Typically, each patient is hospitalized with her/his parents for the entire morning in a dedicated quiet room with bed, table, and chairs, separated from all the other patients. Parents are asked to bring the child’s personal belongings from home which can contribute to making the child more comfortable. The multidisciplinary team of HPs includes child and adolescent neuropsychiatrists, residents in child and adolescent neuropsychiatry, pediatric nurses, psychologists, speech and language therapists, occupational therapists, and EEG technicians.

HPs were asked to quantify the total number of estimated autistic patients met during their professional experience. Their demographics characteristics were collected (i.e., sex, age, and type of profession).

### 2.2. Questionnaire

Each participant HP was asked to respond to a 16-item questionnaire about their personal experience with the patients, using a 5-point Likert scale: for example, participants rated their ability to reassure the patient or the perceived ability to identify the clinical problem despite communication difficulties. The questions are reported in Figure 1. This online survey was spread through WhatsApp among HPs’ colleagues. Participants gave informed consent and completed the survey via an online platform.

### 2.3. Patients with ASD

The data from all consecutively referred children with ASD undergoing a medical diagnostic assessment subsequent to an ASD diagnosis at the University of Turin Pediatric Hospital Regina Margherita Outpatient Service for Neurodevelopmental Disorders between 1 January 2019 and 15 May 2022 were analyzed to ascertain the success in completing the tests. Patients were referred in order to evaluate medical comorbidities or specific etiology (e.g., metabolic, genetic) of ASD. When indicated, patients underwent a blood test, urine test, EEG, ECG, or behavioral audiometry.

Parents gave informed permission to participate, and all the procedures were approved by the local ethics committee.

### 2.4. Evaluation of the Severity of ASD Symptoms

The Autism Diagnostic Observation Schedule-2 (ADOS-2) was used to assess autistic symptoms [28]. This instrument assesses in a semi-structured setting the presence and severity of ASD symptoms, including deficits in communication, social interaction, and play or imaginative use of materials. According to age and language levels (ranging from nonverbal to verbally fluent), appropriate modules are chosen by trained clinicians. In order to measure the severity of ASD symptoms, the total score of each ADOS-2 module was converted into the calibrated severity score (CSS) [29].

### 2.5. Behavioral and Emotional Problems

The parents completed the Child Behavior Checklist 1.5–5 (CBCL), Italian edition [30]. The CBCL is a caregiver report consisting of 100 items identifying emotional and behavioral disorders in children from 1.5 to 5 years old. On the CBCL, behaviors and emotional problems are assessed using a 3-level rating system where 0 represents “not true”, 1 stands for “somewhat/sometimes true”, and 2 indicates “very/often true.” Raw scores are transformed into T-scores, yielding eight syndromic scales, including emotional/reactive, anxious/depressed, somatic complaints, withdrawal, sleep problems, attention problems, aggressive behavior, and other problems. These scales contribute to two broader dimensions, internalization and externalization, as well as a comprehensive score. T-scores are calculated for each scale and for the overall score. T-scores between 65 and 70 are considered to be in the borderline range, while scores above 70 are regarded as clinically abnormal.

Emotional dysregulation (ED) was measured using the Attention, Aggression, and Anxious/Depressed scales of the CBCL (CBCL-AAA). This approach takes into account the affective, behavioral, and cognitive dimensions of ED. The ED score is thus calculated by adding up the CBCL-AAA T scores. The CBCL-AAA was initially used to define Deficient Emotional Self-Regulation (DESR) with scores ranging from 1 to 2 standard deviations (SDs) above the mean. A score above 2 SDs indicates a profile of ED [31]. Previous research has shown that ED strongly contributes to impaired adaptive functioning in young patients with ASD [32].

### 2.6. Statistical Analyses

The statistical programming language R (version 4.0.5) [33] was used to perform the statistical analyses. Descriptive statistics were applied to the sociodemographic and clinical data. Continuous variables were described by mean and SD, and categorical data as percentages. Since the EEG was the most difficult test to obtain results for, we decided to use this variable as an outcome of interest for examining the possible association with demographics and clinical variables. To this end, a binomial regression analysis with Firth’s bias correction [34] for the sample with a complete set of assessments (*n* = 45) was performed, with sex, age, CSS-Tot, and CBCL-AAA as independent variables, and a successfully performed EEG as the dependent variable.

## 3. Results

### 3.1. Sample Characteristics

#### 3.1.1. Health-Care Professionals

Forty-five HPs completed the online survey (54.9% of the total number of HPs working in the unit at the time of the survey). Most participants were female (42, 93.3%); 20 (44.4%) were between 20 and 30 years old, 11 (24.4%) between 31 and 40 years old, and 8 (17.8%) between 41 and 50 years old, and 6 (13.3%) were over the age of 50 years old. Regarding the professional role, the sample consisted of child and adolescent neuropsychiatrists (8.9%), child and adolescent neuropsychiatrists in training (35.6%), nurses (24.5%), speech and language therapists (6.7%), occupational therapists (4.4%), psychologists (8.9%), and EEG technicians (11.1%).

#### 3.1.2. Survey of the Health-Care Professionals

The estimated mean number of children with ASD whom each HP had ever cared for was 90.4 (median 40, SD 165).

A total of 37.2% of the HPs reported aggression as a concern, 37.2% the communication difficulties, and 15.5% the evaluation difficulties (Table 1). Figure 1 shows the responses to the questions about personal attitudes towards autistic patients. All (100%) of the involved HPs also indicated that they would like to receive more training on how to manage children with ASD. Each HP was even questioned about her/his perception of the success rate of the instrumental examinations performed. Of them, 44.4% reported that only 50% of the instrumental examinations had been completed; 28.9% reported a 75% completion rate; and 20% reported a 25% completion rate, while only 2.2% of the HPs reported that 100% of instrumental examinations had been successfully completed.

#### 3.1.3. Patients with ASD

A total of 153 children, 74.5% males, participated in the study, mean age 5.6 years (SD 3.9, range 1–17 years). The full psychometric assessment, comprising CBCL-AAA and ADOS CSS-Tot, was available for 53 patients; of these, 75.5% were male, mean age 3.9 years (SD 2.5).

#### 3.1.4. Completion of the Medical Diagnostic Assessments

The feasibility of the laboratory and instrumental tests depending on the compliance of the children with ASD was evaluated. The success rate was found to be 98.4% for the blood test, 95.9% for the urine test, 91.1% for the behavioral audiometry, and 96.1% for the electrocardiogram. The lowest success rate was found for the EEG (88.5%) (Table 2). In addition, for 21 patients (13.7%), it was necessary to schedule a second admission to the day hospital in order to complete the prescribed tests, since it was not possible to perform all of them in a single day.

In the binomial regression analysis with EEG performance outcome as the dependent variable, no statistically significant relation was found between successful completion and sex, age, CSS-Tot, or CBCL-AAA (Table 3).

## 4. Discussion

This study aimed to investigate the hospital HPs’ perception of their experience in conducting a series of medical and laboratory assessments when caring for young children with ASD in a day hospital setting. The issues that emerged as being of greatest concern to the HPs were the child’s possible aggression and the difficulties in communication. Aggressive behaviors by young patients can be directed against self, others, or objects, and are generally an expression of anger, frustration, or distress. These behavioral problems are indeed more common in children with ASD than in neurotypically developing peers, as reported in a study that found a higher frequency of aggressive behaviors during blood drawing in children and adolescents with ASD compared with normal controls [35]. These behaviors can have an adverse impact on the quality of the care provided and on the associated costs, leading for example to sick leave and injury compensation for the HPs involved [36].

Regarding the difficulties in communication, establishing effective communication with the patient is a main component of clinical practice [37]. However, both receptive communication and expressive language represent a challenge for most children with ASD, albeit with large inter-subject variability. It is estimated that from one-third to one-half of children with ASD have minimal or completely absent functional language [38].

In our sample, all the participating HPs indicated their interest in receiving more training on how to manage children with ASD. A recent review reported that lack of training for HPs is a risk factor for lower quality care for patients with ASD [24]. Providing HPs with tools, resources, and professional training can increase their knowledge and skills, thus leading to greater confidence in managing complicated situations [39].

Moreover, 44.4% of the HPs disagreed and 22.2% strongly disagreed with the statement that the environment was adequate given the needs of children with ASD (Figure 1). Given the sensory hypersensitivity of children with ASD and their reduced capacity to adjust to change, appropriate spaces specifically designed for these patients could help decrease anxiety and discomfort during hospital care [37].

The majority of the HPs in this study were concerned about arriving on time when they have to evaluate a patient with ASD. This is in agreement with available literature, which recommends prioritizing children with ASD and reducing waiting times as much as possible or alternatively setting up a dedicated waiting room provided with toys, clocks, and visual displays [24].

Most of the HPs (55.6%) agreed that the presence and cooperation of parents was “useful,” and 37.8% strongly agreed with this. Parents of children with ASD are indeed an important resource for HPs because of their high level of knowledge of their children’s needs [39]. In many studies, it has been reported that HPs rely on parents not only for their background knowledge of the child, but for their help as active participants during the administration of medical care because of their ability to communicate with the child in the best way [40].

The study also investigated to what extent children with ASD were able to complete the required laboratory and instrumental tests. Contrary to expectations and to the HPs’ perceptions, the non-performance rates of testing due to non-cooperation of the patient with ASD were quite low (between 1.6% and 11.5%). Thus, the rate of failure was apparently overestimated by the HPs, who might have been influenced by the difficulties and inadequacies experienced in caring for children with ASD. In fact, the failure rate of conducting routing medical procedures with ASD children in a dedicated environment proved to be rather small.

The highest completion rate was reported for the blood tests (98.4%), and the lowest rates was found for the EEG (about 11.5%). For several reasons, EEG testing seems to be a particularly complicated and stressful procedure for these patients, with difficulties in tolerating the tactile stimulation from application of the electrodes and from wearing the headset. Indeed, the need for sedation for performing EEG in children with ASD has been reported [41]. Our analyses did not show statistically significant associations between EEG successful completion and clinical characteristics of the children, and this result is consistent with a quasi-randomized trial that evaluated the impact of a blood draw intervention program in ASD [42]. To our knowledge, no other studies had examined the association between clinical variables characterizing ASD and the feasibility of EEG. It is possible that EEG completion may be influenced by other variables than those here examine, such as the parent’s ability to manage their children in difficult contexts or the HPs’ skills and experience with caring for ASD patients.

It must be considered that the examinations and procedure here described were performed at a day hospital service for neurodevelopmental disorders. This is a specialized environment for children with ASD. The surveyed HPs have experience in working with children who have neurological and neurodevelopmental disorders. In addition, the setting is appropriate for these patients, with a special room dedicated to them throughout their stay, so that sensory stimuli can be minimized. Nonetheless, to further alleviate the stress associated with medical procedures for the children, their parents, and HPs, a series of initiatives have recently been introduced in daily practice, in accordance with the recommended best practices and guidelines [25]. These include an easy-read version of the hospital procedure with photos that the family can use with the child and a pre-assessment day to allow the children and their family to familiarize themselves with the day hospital environment. It will be interesting to evaluate the improvements associated with such practices in future research.

This study has several limitations. The sample of HPs who responded to the questionnaire had diverse professional roles and experiences with ASD, the number of patients encountered by each professional was heterogeneous, the number of HPs was relatively small, and the data came from only one university hospital, thus limiting the possibility of generalizing the results. Another limitation is that we did not collect data on a control sample of patients with other conditions (such as ADHD, epilepsy, intellectual disability without ASD); thus, this study design cannot answer the question of whether ASD children are more challenging than children with other conditions. Finally, the secondary analysis on clinical features which may possibly influence tests’ feasibility was limited to a small part of the whole sample.

This study details HPs’ perceptions about the management of ASD patients in a day hospital setting. Despite reports of great challenges, very few assessments could not be performed for lack of collaboration by the children, and the completion rate was not related to the severity of the ASD symptoms, suggesting that other factors may be at play in influencing the feasibility of medical assessments in ASD children.

## Figures and Tables

**Figure 1 children-10-00476-f001:**
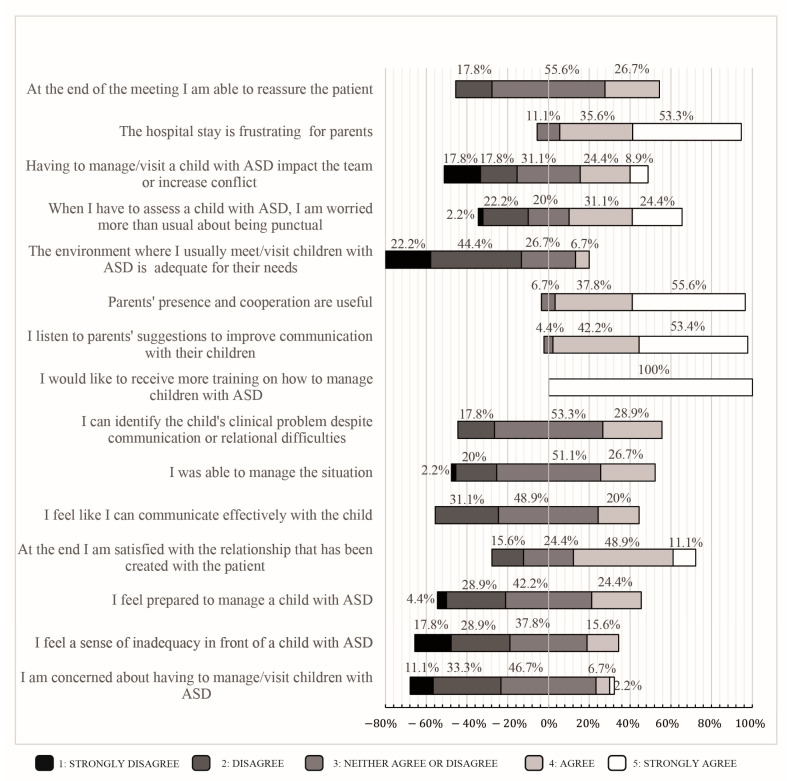
HP view of their interaction with ASD children and their families (*n* = 45). The questionnaire items on a 5-point Likert scale, with answers provided as percentages.

**Table 1 children-10-00476-t001:** Main concerns raised by HPs when taking care of patients with ASD (*n* = 45).

Concern	*n* (%)
Aggression	16 (37.2%)
Child and/or caregiver compliance	6 (13.3%)
Communication difficulties	16 (37.2%)
Evaluation difficulties	7 (15.5%)
Need for more training	45 (100%)

**Table 2 children-10-00476-t002:** Completion rate of laboratory and instrumental tests in children with ASD.

Test	Attempted *n* *	Successfully Completed *n* %
Blood tests	123	121, 98.4
Urine tests	97	93, 95.9
Behavioral audiometry	45	41, 91.1
Electrocardiogram	77	74, 96.1
Electroencephalogram	130	115, 88.5

* Variable *n* since not all the laboratory and instrumental tests were requested for every patient.

**Table 3 children-10-00476-t003:** Results of a binomial regression analysis testing patient sex, age, severity of ASD (CSS-Tot) and of emotional dysregulation (CBCL-AAA) as factors influencing the successful completion of EEG (*n* = 45) *.

Variable	Odds Ratios	CI	*p* Value
Sex	1.57	0.11–22.10	0.738
Age	1.07	0.64–1.80	0.786
CSS-Tot	0.83	0.47–1.46	0.516
CBCL-AAA	1.01	0.95–1.07	0.804

* Included are the children undergoing the EEG procedure for whom the full psychometric assessments were available (*n* = 45). CBCL-AAA: Child Behavior Check-List—Attention, Aggression, and Anxious/Depressed scales; CSS-Tot: calibrated severity score of ASD—Total.

## Data Availability

Requests for the study data can be addressed to the authors. The raw data are not publicly available due to privacy considerations.

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
