# Peer review of "Working with Children with Autism Undergoing Health-Care Assessments in a Day Hospital Setting: A Perspective from the Health-Care Professionals"

_children, 2023, doi:10.3390/children10030476_

Round 1
Reviewer 1 Report
The authors conducted a very interesting study which deserves to be expanded at national level.
I find very interesting the data discussed in lines 267-268 "Contrary to expectations and health professional perceptions, the non-performance rates of testing due to non-cooperation of the patient with ASD were quite low (between 1.6% and 11.5%)." I suggest that the Authors discuss a possible explanation of this fact.
In line 185 there is a question mark to remove: (4.4%), psychologists (8.9%?) and EEG technicians (11.1%).
Author Response
REVIEWER 1
The authors conducted a very interesting study which deserves to be expanded at national level.
I find very interesting the data discussed in lines 267-268 "Contrary to expectations and health professional perceptions, the non-performance rates of testing due to non-cooperation of the patient with ASD were quite low (between 1.6% and 11.5%)." I suggest that the Authors discuss a possible explanation of this fact.
Thank you for this suggestion: we have expanded the discussion by commenting on possible explanations of this finding.
In line 185 there is a question mark to remove: (4.4%), psychologists (8.9%?) and EEG technicians (11.1%).
We thank the Reviewer for highlighting this typo. We have corrected the text by removing the question mark.
Reviewer 2 Report
Thank you for providing me with an opportunity to review this manuscript. I found it fascinating. However, some could be clearer to me.
Introduction
- The authors should provide a bigger picture of the challenge of conducting tests among children in the non-ASD group. For example, some tests are obnoxious and cause anxiety and noncooperation to any child, not just only ASD.
- Is this the first survey among healthcare providers? If not, please mention other related research.
- What is HCW (line 125)? ED(line 158)?
- The title of 2.1.5. Evaluation of Emotional Dysregulation should be "behavioural and emotional problems."
- What is the ratio of participation of the healthcare providers in the hospital?
- What is this question mark for? - psychologists (8.9%?) (line 185)
Results
7. 37.2% of the health care professionals reported aggression as a concern, and 51.5% 189 the communication difficulties- These results are from Figure 1 or somewhere else. Can the authors present it in a table if they are not from Figure 1?
8. Did only 53 complete all the diagnostic assessments?- That's only 34% ( line 201). Judging from Table 1, those who could complete all the tests should be more than 34%.
Table 2. Results of a binomial regression analysis testing patient sex, age, severity of ASD (CSS-Tot) 217 and emotional dysregulation (CBCL-AAA) as factors influencing the successful completion of 218 EEG (n=53)*.
Table 2 is confusing. What is the total number for this analysis of completeness; 153?
9. 53 (Yes/ complete) vs. 100 (No/incomplete). More details and clear procedures of analysis should be described in the analysis part.
- Please provide N for figure 1
- Correlation analysis between Behavioral/ emotional problems and completion is more valuable than regression because the sample is small. In addition, it is worth looking at the correlation between each syndrome.
Discussion
12. Line 233; This statement is unclear- In a sample of 74 children and adolescents with ASD and 115 typically developing control, the frequency of aggressive behaviours during blood drawing was significantly higher in ASD [27].
- Line 243. Are there any results the authors referred to in this study? – not clear ( Also see Q7)
- The same as in line 249, ' environment'. Are these results from Figure 1?
- The same as in line 254, 'arriving on time'. Are these results from Figure 1?
- Line 259-260, ……..quite useful," and 37.8 % that it was "very" helpful.
17. Is the response option different for this item? They seem to be "agree–disagree" responses.
References
18. Please check ref no. 8- it is not English.
Author Response
REVIEWER 2
Thank you for providing me with an opportunity to review this manuscript. I found it fascinating. However, some could be clearer to me.
Introduction
- The authors should provide a bigger picture of the challenge of conducting tests among children in the non-ASD group. For example, some tests are obnoxious and cause anxiety and noncooperation to any child, not just only ASD.
Thank you for pointing this out. A new paragraph was added to the Introduction section synthesizing how stressful medical procedures generally are in children and which factors can influence the stress.
- Is this the first survey among healthcare providers? If not, please mention other related research.
We thank the reviewer for this question. In response, we have included a paragraph about previous surveys referencing a recent literature review in the manuscript. Previous surveys among healthcare professionals have reported only moderate levels of self-efficacy, insufficient knowledge about autism and lack of relevant training.
- What is HCW (line 125)? ED(line 158)?
Thank you for pointing out the inconsistencies in acronym use. In the revised manuscript, we have replaced HCW (health care workers) with HP throughout the manuscript, specifying it in full (healthcare professional) at its first occurrence in the text. The same was done for ED, which stands for emotional dysregulation: ED is written out in full at its first occurrence in the text.
- The title of 2.1.5. Evaluation of Emotional Dysregulation should be "behavioural and emotional problems."
We thank the Reviewer for the suggestion. The title of 2.1.5 was modified accordingly.
- What is the ratio of participation of the healthcare providers in the hospital? Total HP number of HP of the Division of Child and Adolescent Neuropsychiatry of the University of Turin ( Pediatric Hospital Regina Margherita) is 82, 45 HP participated, so the proportion is 54.9%. This percentage was included in the manuscript.
- What is this question mark for? - psychologists (8.9%?) (line 185)
We thank the Reviewer for highlighting this typo. We have deleted the question mark.
Results
- 37.2% of the health care professionals reported aggression as a concern, and 51.5% 189 the communication difficulties- These results are from Figure 1 or somewhere else. Can the authors present it in a table if they are not from Figure 1?
Thank you for raising this issue. A table describing the main concerns raised by the HPs was included in the manuscript (Table 1).
- Did only 53 complete all the diagnostic assessments?- That's only 34% ( line 201). Judging from Table 1, those who could complete all the tests should be more than 34%.
Thank you for pointing out this issue and letting us clarify it. A psychometric assessment comprising CBCL-AAA and ADOS CSS-Tot was available for 53 patients. This figure is different from the total number of children requiring laboratory and/or instrumental tests (n=153). We clarified the sample size in the manuscript according to your suggestion.
Table 2. Results of a binomial regression analysis testing patient sex, age, severity of ASD (CSS-Tot) 217 and emotional dysregulation (CBCL-AAA) as factors influencing the successful completion of 218 EEG (n=53)*.
Table 2 is confusing. What is the total number for this analysis of completeness; 153?
- 53 (Yes/ complete) vs. 100 (No/incomplete). More details and clear procedures of analysis should be described in the analysis part.
Thank you for pointing this out. We have clarified the sample size of the analyses reported in Table 3. Both ADOS CSS-Tot and CBCL AAA index were available for 45 of the children who underwent the EEG precedure. The regression analysis evaluated possible factors associated with incomplete EEG in this subgroup (n=45).
- Please provide N for figure 1.
We have provided the N in the figure as requested.
- Correlation analysis between Behavioral/ emotional problems and completion is more valuable than regression because the sample is small. In addition, it is worth looking at the correlation between each syndrome.
We agree with the reviewer that performing statistical inference with a small samples size can be problematic. We chose to perform a binomial regression with Firth’s bias correction (Firth, 1993) and a small number of regressors, as we believe this is one of the most robust techniques under our conditions . Nonetheless, according to the reviewer's suggestion, we have also performed a Mann-Whitney test to compare differences between groups (successful vs unsuccessful EEG) with regard to ADOS CSS-Tot (p=1), CBCL AAA index (p=0.35) and age (p=0.38). There was no difference in the distribution of sex in the two groups.
Discussion
- Line 233; This statement is unclear- In a sample of 74 children and adolescents with ASD and 115 typically developing control, the frequency of aggressive behaviours during blood drawing was significantly higher in ASD [27].
- We have clarified the sentence in the text.
- Line 243. Are there any results the authors referred to in this study? – not clear ( Also see Q7)
We added this results in the results section (3.1.2. Survey of the Healthcare Professionals): “100% of the involved professionals also responded that they would like to receive more training regarding how to manage children with ASD.”
- The same as in line 249, ' environment'. Are these results from Figure 1?
Yes, we have added the reference to figure 1 in the manuscript.
- The same as in line 254, 'arriving on time'. Are these results from Figure 1?
Yes, we have specified this in the revised text.
- Line 259-260, ……..quite useful," and 37.8 % that it was "very" helpful.
- Is the response option different for this item? They seem to be "agree–disagree" responses.
Thank you for pointing this out. The response options were the same. We have corrected the manuscript accordingly.
References
- Please check ref no. 8- it is not English.
Thank you, we changed the reference with an English one.
Reviewer 3 Report
Thank you for the opportunity to review the manuscript regarding the perspective of health-care professionals in a day hospital setting supporting children with autism and their families when they attend for health-care testing. It is an important manuscript because children with autism should be included and be able to access mainstream health and medical services.
The study design is appropriate and its findings are adequately discussed. The manuscript is well written for an international audience.
In the Discussion, it would be useful for the authors to offer further recommendations to alleviate the stress and anxiety of children with autism and their families, such as an Easy Read version of the hospital procedure with photos or videoclips that the family can use with the child and or a pre-assessment day for the children and their family to be able to attend the day hospital so as to familiarise themselves with the clinical environment. These suggestions are not new and have been used in other countries.
I suggest the authors reconsider the title of the manuscript to reflect a more inclusive approach to supporting children with disabilities. A title for consideration might be "Working with children with autism undergoing health-care assessments in a day hospital setting: A perspective of health-care professionals."
Author Response
REVIEWER 3
Thank you for the opportunity to review the manuscript regarding the perspective of health-care professionals in a day hospital setting supporting children with autism and their families when they attend for health-care testing. It is an important manuscript because children with autism should be included and be able to access mainstream health and medical services.
The study design is appropriate and its findings are adequately discussed. The manuscript is well written for an international audience.
In the Discussion, it would be useful for the authors to offer further recommendations to alleviate the stress and anxiety of children with autism and their families, such as an Easy Read version of the hospital procedure with photos or videoclips that the family can use with the child and or a pre-assessment day for the children and their family to be able to attend the day hospital so as to familiarise themselves with the clinical environment. These suggestions are not new and have been used in other countries.
Thank you for pointing this out. We have recently implemented a series of practices in our clinic, such as an Easy Read version of the hospital procedure with photos that the family can use with the child and a pre-assessment visit to allow the children and their family to familiarize themselves with the day-hospital environment. We described such practices in the discussion.
I suggest the authors reconsider the title of the manuscript to reflect a more inclusive approach to supporting children with disabilities. A title for consideration might be "Working with children with autism undergoing health-care assessments in a day hospital setting: A perspective of health-care professionals."
Thank you for your suggestion, we changed the title accordingly, in order to have a more inclusive approach.
Round 2
Reviewer 2 Report
Your revised version is considerably improved.
I have no further concerns. Congratulations!